# ADVERSARIAL EXAMPLES FOR HEURISTICS IN COMBINATORIAL OPTIMIZATION:
# AN LLM BASED APPROACH

## ABSTRACT

This work employs LLMs to generate adversarial examples for heuristics in combinatorial optimization. The problem, given a heuristic for an optimization problem, is to generate a problem instance where the heuristic performs poorly. We find improved adversarial constructions for well-known heuristics for k-median clustering, bin packing, the knapsack problem, and a generalization of Lovász's gasoline problem. Specifically, we adapt the FunSearch framework [Romera-Paredes et al., Nature 2023] to obtain adversarial constructions for these problems. We note that using FunSearch is crucial to our improved constructions — local search does not give comparable results. The advantage of FunSearch is that it produces structured instances that yield theoretical insights which are post-processed and generalized by a human researcher, while other metaheuristics usually produce only unstructured instances that are harder to generalize.

## 1 INTRODUCTION

Recent advances in neural networks, particularly large language models, have provided a new impetus to the use of artificial intelligence (AI) in all areas of science. Mathematics is of particular interest to AI researchers due to its formalism, perfect verification ability, being an open domain as well as being a domain which is a hallmark of reasoning and intelligence. In the last decade, AI made advances in research by proposing new conjectures in niche domains such as knot theory (Davies et al., 2021), discovered new algorithms (Fawzi et al., 2022) and provided new lower bounds or combinatorial constructions (Romera-Paredes et al., 2024; Novikov et al., 2025; Wagner, 2021; Mehrabian et al., 2023).

Some of the works (Fawzi et al., 2022; Wagner, 2021; Mehrabian et al., 2023) use black-box neural networks to make these advances, which provide new discoveries yet provide almost no interpretation or insight to mathematicians to advance the field itself. The use of large language models in works like (Romera-Paredes et al., 2024; Novikov et al., 2025) provided an opportunity to bridge this gap by using natural language or coding languages as an interface of communication between mathematicians and black-box neural networks. In this work, we take this inspiration forward to generate adversarial examples for heuristics used in combinatorial problems. Many combinatorial optimization problems have widespread real-world applications but are computationally intractable (e.g., NP-hard). A natural way to solve these problems in practice is to devise heuristics. Ideally, we aim to analyze a heuristic to guarantee an upper bound on its approximation ratio and provide a matching lower bound by constructing an example where the heuristic achieves this bound. However, this is often not achievable in practice. Instead, one typically tries to either tighten the upper bound through more refined analysis or raise the lower bound by identifying a better adversarial example.

In this work, we use the FunSearch paradigm (proposed by Romera-Paredes et al. (2024)) to generate adversarial examples for heuristics to solve a variety of combinatorial optimization problems. Analyzing the worst-case performance of heuristics can explain their performance in real-world applications, and knowledge of adversarial instances can help devise better heuristics. Specifically, we consider well-known heuristics for the knapsack problem, bin packing, hierarchical clustering, and a variant of Lovász's gasoline puzzle. This approach of using FunSearch in contrast to local search has several advantages: a) Local search algorithms which search in vector spaces only find one vector.

FunSearch can find a Python-program that is generic on parameters like the dimension or size of the instance, and thus generalizes. Secondly, the result of local search, just being a vector of unlabeled numbers, does not lend itself to interpretation. The result of FunSearch is (usually short) Python-code that can be interpreted and modified by humans. Thirdly, for many optimization-problems, optimal solutions have a lot of structure and symmetry, i.e. low Kolmogorov-complexity. Simple local search encodes all vectors indifferently and does not account for symmetries, while FunSearch does. The most striking example was for bin packing, where local search provided comparable scores for small numbers of bins. Unfortunately, the solution found by local search did not follow a discernible pattern[1], while the code from FunSearch (see Fig. 2b) was generalized to obtain a lower bound of 1.5 (whereas the previous best-known lower bound was 1.3). We refer to Section 2.2.2 and Fig. 2b for further details.

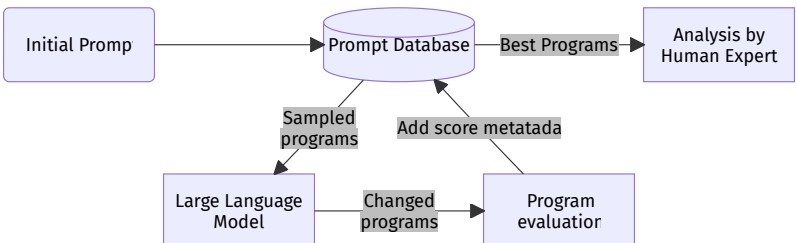

Figure 1: A diagrammatic representation of Co-FunSearch.

We start with adversarial examples provided to FunSearch, and then inspect the programs which yielded the highest scores. Some of these programs might have a discernible structure relevant to the problem, while others consist of hard-coded or pseudo-random numbers which expose no useful structure. We then manually inspect and modify the promising programs: We eliminate parts whose removal does not decrease the score, and we modify the program to be simpler where possible. For example, this may involve removing redundant elements of lists (Fig. 2, Fig. 5), or simplifying a list of $n$ ascending numbers into a list containing the mean of those numbers $n$ times (Fig. 5). Afterward, we attempt to prove statements about the scores of the instances, or otherwise feed the simplified programs back into FunSearch to obtain better results. These expert modifications were essential for generating meaningful insights, and the collaborative workflow demonstrates FunSearch's potential for productive partnerships between mathematicians and AI systems. We call this method Co-FunSearch: *Collaborative FunSearch*.

Particularly, with Co-FunSearch, we were able to *disprove* that the Nemhauser-Ullmann heuristic for knapsack problem has output-polynomial running time, and we improve the lower bound of best fit heuristic for bin packing in the random order model from 1.3 to 1.5. We also obtained the *first non-trivial lower bound of the golden ratio* for the price of hierarchy for $k$-median clustering, and *disprove the conjecture* that the iterative rounding algorithm for the generalized gasoline problem is a 2-approximation.

We conclude with some challenges and limitations of our approach in Section 4.

## 2 PROBLEMS AND NOTATION

### 2.1 GENERAL FRAMEWORK FOR ADVERSARIAL INSTANCE GENERATION

We first propose a general framework for generating adversarial instances for any given heuristic, and then describe the particular problems we focus on in this work and how we instantiate this general framework for the given problem. Given an optimization problem (without loss of generality, a minimization problem), a heuristic algorithm $\mathcal{H}$ and a (computationally expensive) optimal algorithm *Opt*, the goal is to construct an instance $\mathcal{I}$ where the heuristic performs poorly with respect to *Opt*. More concretely for minimization problems, we aim to construct an adversarial instance $\mathcal{I}$

---

[1]For instance, one of the local-search-generated lists outperforming FunSearch was: [0.003031, 0.005466, 0.006098, 0.007283, 0.021158, 0.068030, 0.073417, 0.170490, 0.202092, 0.219287, 0.306771, 0.375912, 0.540358].

such that the ratio $R = \frac{\text{Score}(\mathcal{H}(\mathcal{I}))}{\text{Score}(Opt(\mathcal{I}))}$ is large, where $\text{Score}(\mathcal{H}(\mathcal{I}))$ denotes the value yielded by the heuristic algorithm and $\text{Score}(Opt(\mathcal{I}))$ denotes the optimum value for $\mathcal{I}$.

While methods like local search, tabu search, and genetic algorithms have focused on generating adversarial instances for heuristics, this work focuses on using language models for generating the instances. Specifically, we model each instance as output of a program $\mathcal{P}$ s.t. $\mathcal{I} = \text{Output}(\mathcal{P})$. Initially, a trivial instance is expressed as program $\mathcal{P}_0$. In addition, we prompt a large language model $\mathcal{L}$ that has proficiency in code generation and reasoning. At each iteration $i$, the language model takes as input one of the previously generated programs, $p = \mathcal{P}_{<i}$ and generates an improved version $p'$ of $p$ such that it improves the reward $R$. We specifically follow the evolutionary approach used in Romera-Paredes et al. (2024) for generating these programs and optimizing the reward $R$.

## 2.2 PROBLEMS AND HEURISTICS

We focus on four distinct problems and their corresponding heuristics to illustrate the effectiveness of this approach. These problems vary from knapsack, bin-packing, hierarchical clustering to the gasoline puzzle by Lovász. While the approach is general, we believe the specific instantiation on these problems provides a general lens to find adversarial instances for any given heuristic.

### 2.2.1 NEMHAUSER-ULLMANN HEURISTIC FOR THE KNAPSACK PROBLEM

In the classical NP-hard knapsack problem, an input consists of a set of $n$ items, where each item $i \in [n]$ has a profit $p_i \in \mathbb{R}_{>0}$ and a weight $w_i \in \mathbb{R}_{>0}$. Additionally, a capacity $t \in \mathbb{R}_{>0}$ is given, and the goal is to find a subset $I \subseteq [n]$ of the items such that the profit $\sum_{i \in I} p_i$ is maximized under the constraint $\sum_{i \in I} w_i \leq t$. Without a given capacity $t$, the knapsack problem can also be viewed as a bi-objective optimization problem, where one wants to find a subset with small weight and large profit. These two objectives are obviously conflicting and there is no clear optimal solution anymore, but one rather has to find a good trade-off between the criteria. In multi-objective optimization, it is very common to compute the set of Pareto-optimal solutions where a solution is called Pareto-optimal if there does not exist another solution that is simultaneously better in all objectives (see, e.g., Ehrgott (2005) for a comprehensive overview). Only Pareto-optimal solutions constitute reasonable trade-offs and for many multi-objective optimization problems, algorithms for computing the set of Pareto-optimal solutions are known (e.g., for the multi-objective shortest path problem (Corley & Moon, 1985)). These are usually no polynomial-time algorithms, as the set of Pareto-optimal solutions can be of exponential size. However, in practice the Pareto set is often small and one is interested in finding algorithms that are output-polynomial time, i.e., whose running time depends polynomially on the input and the output size. Such algorithms are efficient if the Pareto set is small, which is often the case in applications.

**Nemhauser-Ullmann heuristic** It is an open problem whether output-polynomial time algorithms for the knapsack problem (viewed as a bi-objective optimization problem) exist (Röglin, 2020). The best candidate for such an algorithm is the Nemhauser-Ullmann algorithm, which is based on dynamic programming (Nemhauser & Ullmann, 1969). For a given instance of the knapsack problem with $n$ items, it computes iteratively the Pareto sets $\mathcal{P}_1, \ldots, \mathcal{P}_n$, where $\mathcal{P}_i$ denotes the Pareto set of the sub-instance that consists only of the first $i$ items (i.e., $\mathcal{P}_n$ is the Pareto set of the entire instance). The Nemhauser-Ullmann algorithm can be implemented to run in time $O(\sum_{i=1}^{n} |\mathcal{P}_i|)$. If there was an $\alpha$ such that $|\mathcal{P}_i| \leq \alpha|\mathcal{P}_n|$ for each instance and each $i$, one could bound the running time by $O(\alpha n |\mathcal{P}_n|)$, which would result in an output-polynomial time algorithm as long as $\alpha$ grows at most polynomially with $n$. So far, no instances were known where an intermediate set $\mathcal{P}_i$ is larger than the final Pareto set $\mathcal{P}_n$ by more than a small constant factor. With the help of an instance generated by FunSearch, we construct a sequence of instances disproving that the Nemhauser-Ullmann algorithm has output-polynomial running time.

### 2.2.2 BEST FIT HEURISTIC FOR BIN PACKING

Bin Packing is a classical NP-hard optimization problem that has been studied extensively as an online problem. In this problem, items with sizes $w_1, w_2, w_3, \ldots$ arrive one by one and an online algorithm has to assign each item irrevocably to a bin when it arrives. There is an unlimited number of bins with a fixed capacity $c$ available. The goal is to use as few bins as possible to pack all items.

In the online setting, simple algorithms like First-Fit and Best-Fit have been studied, which pack each arriving item into the first bin into which it fits or the fullest bin into which it fits, respectively. To mitigate the power of the adversary in classical worst-case analysis, these algorithms have been studied extensively in the random order setting, in which an adversary chooses the items sizes but the items arrive in a random order. In the unshuffled setting, Dósa & Sgall (2014) proved an upper bound of 1.7 on the approximation-ratio of Best-Fit. This means that, on any instance, the expected number of bins used by Best-Fit is at most 1.7 times the optimal number. As this holds for any instance, this upper bound also applies to the shuffled setting. In the shuffled setting, the best-known lower bound was 1.3, i.e. there exists an instance such that, when the instance is shuffled, Best-Fit needs at least 1.3 times the optimal number of bins, in expectation (Albers et al., 2021). With the help of FunSearch, we improve this lower bound to 1.5.

### 2.2.3 k-MEDIAN IN HIERARCHICAL CLUSTERING

Hierarchical clustering is an important research topic in unsupervised learning. In such a clustering problem, usually a data set $X$ with $n$ points is given and one seeks for a sequence $\mathcal{H}_1, \ldots, \mathcal{H}_n$ of clusterings, where each $\mathcal{H}_k$ is a $k$-clustering of $X$, i.e. a partition of $X$ into $k$ parts. The clusterings must be hierarchically compatible, meaning that each $\mathcal{H}_k$ is obtained from $\mathcal{H}_{k+1}$ by merging two clusters. To evaluate the quality of such a hierarchical clustering, a common approach is to choose an objective function $\Phi$ like $k$-center, $k$-median, or $k$-means and to compare each clustering $\mathcal{H}_k$ with an optimal $k$-clustering $\mathrm{OPT}_k$ with respect to the objective $\Phi$. Then the approximation factor $\alpha$ of the hierarchical clustering can be defined as the worst approximation factor of any of the levels, i.e., $\alpha = \max_{k \in [n]} \Phi(\mathcal{H}_k)/\Phi(\mathrm{OPT}_k)$ (see, e.g., Lin et al. (2010)). Since the optimal clusterings are usually not hierarchically compatible, an approximation factor of 1 cannot be achieved even with unlimited running time. Arutyunova & Röglin (2025) defined the *price of hierarchy* of a clustering objective $\Phi$ as the best approximation factor that can be achieved for any clustering instance. They showed, e.g., that the price of hierarchy for the $k$-center objective is exactly 4, meaning that for any instance of the hierarchical $k$-center problem there exists a hierarchical clustering with an approximation factor of 4 and that there exists an instance for which any hierarchical clustering does not have a better approximation factor than 4. For the $k$-median problem, no non-trivial lower bound on the price of hierarchy is known. The best known upper bound is 16 for general metrics (Dai, 2014). We obtain the first non-trivial lower bound for the price of hierarchy for the $k$-median problem, showing that it is at least the golden ratio, $\approx 1.618$.

### 2.2.4 GASOLINE PROBLEM

The Gasoline problem is a combinatorial optimization problem inspired by Lovász's gasoline puzzle (Lovász, 2007). In an instance of this problem, we are given two sets $X = \{x_1, \ldots, x_n\}$ and $Y = \{y_1, \ldots, y_n\}$ of non-negative numbers with the same sum. The goal is to find a permutation $\pi$ of the set $X$ that minimizes the value of $\eta$ such that

$$\forall [k, \ell]: \quad \left| \sum_{i \in [k,\ell]} x_{\pi(i)} - \sum_{i \in [k,\ell-1]} y_i \right| \le \eta.$$

Given a circle with $n$ points labeled 1 through $n$, the interval $[k, \ell]$ denotes a consecutive subset of integers assigned to points $k$ through $\ell$. For example, $[5, 8] = \{5, 6, 7, 8\}$, and $[n - 1, 3] = \{n - 1, n, 1, 2, 3\}$. The intuition is that the $y_i$-values correspond to road segments on a cycle and the $x_i$-values correspond to fuel canisters that can be placed between the segments. The goal is to distribute the canisters such that one can get around the cycle with the smallest possible fuel tank capacity $\eta$.

The Gasoline problem is known to be NP-hard, and a 2-approximation algorithm for it is known (Newman et al., 2018). It is an open problem whether better approximation algorithms or even a polynomial-time approximation scheme exist. In the literature, another heuristic for the problem has been considered that is based on iteratively rounding the linear programming relaxation (Rajković, 2022). The approximation guarantee of this algorithm is unknown. In his master's thesis, Lorieau constructed a class of instances showing that its approximation factor is not better than 2 (Lorieau, 2024). Lorieau conjectured that it is actually a 2-approximation algorithm, but this has not been proven yet. The iterative rounding algorithm is interesting because it generalizes canonically to a $d$-dimensional Gasoline problem in which $x_i$ and $y_i$ are $d$-dimensional vectors. Also for this

generalization, the best-known lower bound was 2 and Lorieau conjectured that also for this generalization the algorithm achieves a 2-approximation. With Co-FunSearch, we obtain a family of instances disproving this conjecture.

## 3  EXPERIMENTAL DETAILS AND RESULTS

We compare Co-FunSearch to base FunSearch and local search on the above 4 problems. The main goal in all these problems is to search for a vector $v$ (encoding the instance) which optimizes the given objective (usually some performance-measure of some heuristic on this specific instance). The objectives depend on the problem, and are detailed below in section 3.3. Random search works by initializing a random vector $v$. At each step, sample a random vector $v'$ close to $v$ and check if $v'$ improves on the objective. If it does, replace $v$ by $v'$ with some probability $p$, otherwise keep $v$ unchanged. This procedure keeps improving on the objective until reaching a local minimum. For our experiments, $v'$ arises from $v$ by adding independent normally-distributed noise with mean 0 and variance $s \cdot (1 - \frac{t}{t_{\max}})$ to each coordinate of $v$ (clipping $v'$ to the problem's bounds as required), where $t$ is the current time since the start of the search, $t_{\max}$ is the time after which we terminate the search (set to 3 minutes), and $s$ is a problem-specific parameter. For the knapsack-problem, we chose 20 items and $s = 1000$, because both weights and profits were rounded before evaluation to be less sensitive to floating-point imprecision. For bin-packing, we chose 13 bins with capacity 1 and $s = 0.2$. For weighted hierarchical clustering, we chose 8 points, $s = 0.2$, and replaced each point's weight $w$ to $2^w$ before evaluation, because we observed worst-case instances' weights frequently spanning several orders of magnitude. For the two-dimensional gasoline-problem, we chose $s = 0.2$ and $|X| = |Y| = 14$.

FunSearch works similarly: Instead of searching for a vector $v$ that has a high objective, it searches for a Python-program $P$ outputting a vector with high objective. Sampling a Python-program $P'$ "close" to $P$ is not done by randomly changing characters in the program's source-code, but by prompting an LLM with the source-code of $P$, requesting a similar program which improves the score. The scoring-function is not provided to the LLM. The newly generated program (if it executes without error) is added to a database of programs with its score. In the next iteration, a new program is sampled from the database according to a probability distribution and the process is repeated. More details about the evolutionary search can be found in Romera-Paredes et al. (2024). To evaluate a given program, we use problem-specific scoring-functions, described in their respective sections below.

### 3.1  RESULTS

| Method | Knapsack | Bin-Packing | $k$-median | Gasoline |
|---|---|---|---|---|
| Previous Best Known Lower Bound | 2.0 | 1.3 | 1.0 | 2.0 |
| Local Search | 1.93 | 1.478 | 1.36 | 2.11 |
| FunSearch | 646.92 | 1.497 | 1.538 | 3.05 |
| Co-FunSearch | $n^{O(\sqrt{n})}$ | 1.5 | 1.618 | 4.65 |
| Known Upper Bound | $O(2^n)$ | 1.7 | 16 | None |

Table 1: Comparison of Co-FunSearch with base FunSearch, local search and SOTA on different problems. The given values for local search and FunSearch are the maxima across 30 trials each.

Table 1 outlines the main results for all four problems. Our main results are as follows:

- For the knapsack problem, the local search method only achieves 1.93, whereas FunSearch found instances with a score of 646.92. The compact program found by FunSearch could be improved to a general super-polynomial bound $n^{O(\sqrt{n})}$.
- For the Best-Fit heuristic for bin packing, FunSearch finds an instance which is 1.497 times worse than optimal, outperforming both the existing SOTA (1.3) and local search (1.478). This instance could easily be generalized, yielding an asymptotic bound of 1.5.
- For the hierarchical $k$-median problem, no non-trivial lower bounds were previously known. FunSearch (1.538) outperforms local search (1.36) with an instance that we could modify to yield a lower bound of the golden ratio ($\approx 1.618$).

- Lastly, in Lovász's gasoline problem, FunSearch (3.05) outperforms both the SOTA (2.0) and local search (2.11), and could be further improved to 4.65.

**Generated Programs with FunSearch and Co-FunSearch**   In this section, we illustrate the programs found by FunSearch and how these programs are modified by experts to obtain adversarial instances which are much better in score and are generalizable with guarantees. Fig. 2a shows the initial program given in the bin-packing problem, Fig. 2b shows the instance generated by FunSearch, which achieves a score of $1.4978$, and Fig. 2c shows how we generalized this instance: The instance consists of two types of items in a list which are generalized as entries "$a$" and "$b$" in the figure. Specifically, for large $a$ and $b$, this instance's score approaches $1.5$. Similar Fig. 5 and 6 are shown for hierarchical clustering and the gasoline problem respectively in the appendix.

```python
def get_items() -> list[float]:
    """Return a new bin-packing-instance,
    ↪    specified by the list of items.

    The items must be floats between 0 and
    ↪    1."""
    items = [0.4, 0.5, 0.6]
    return items
```

```python
def get_items() -> list[float]:
    a = 7
    b = 5
    return [1.0 / a] * a + [1.0 / b] * b
```

(a) Initial program.

(c) An intermittent step of tuning 2b by hand

```python
def get_items() -> list[float]:
    """Return a new bin-packing-instance, specified by the list of items.

    The items must be floats between 0 and 1."""
    """Yet another version of `get_items_v0`, `get_items_v1`, and `get_items_v2`, with some
    ↪    lines altered."""
    items = [0.8, 0.2, 0.6, 0.4]
    # Split the first item into seven smaller items and the fourth item into five smaller
    ↪    items
    items = [0.114, 0.114, 0.114, 0.114, 0.114, 0.114, 0.114] + items[1:3] + [0.08, 0.08,
    ↪    0.08, 0.08, 0.08]
    return items
```

(b) A program found by FunSearch after 10 trials of 2,400 samples each.

Figure 2: The evolution of programs generating bin packing instances, with model open-mistral-nemo and a temperature of $1.5$.

## 3.2   ABLATIONS

Figure 3 shows the search dynamics with variations across different models, the temperature parameter and the initial program used during FunSearch. In all these experiments, we plot the maximum score of samples produced so far against the number of samples (LLM-prompts), together with the standard error across 30 trials. To illustrate the effect of variations and due to high computational cost (inference costs) of each experiment, we undertake these ablations on a single problem but believe similar trends would hold for all the other problems as well.

**Variations across different models:** Fig. 3a shows the variations with two models from OpenAI, gpt-4.1-mini (OpenAI, 2025a) and gpt-4.1-nano (OpenAI, 2025b) with Mistral AI's open-mistral-nemo model (Mistral AI, 2024). We observe that gpt-4.1-nano slightly outperforms gpt-4.1-mini. This is a bit counterintuitive, as gpt-4.1-mini is a more powerful model than gpt-4.1-nano. To investigate this further, we plot the both the maximum score and the rolling average score of the last 10 samples (Figure 3b). Here, gpt-4.1-mini outperforms gpt-4.1-nano on the rolling average but performs slightly poorer on the maximum score, highlighting that, although larger models are stronger on average, in problems with verifiable score where one cares about the best performing sample, smaller models are sufficient and can outperform larger ones.

**Variations across temperature:** Fig. 3c shows the variations of the objective function with the change in the sampling temperature. The sampling temperature is an indicator of sharpness of the LLM's probability distribution for each sample (the lower the temperature, the more sharp it is). We observe that the higher sampling temperature performs better than lower sampling temperature, owing to high entropy of samples produced in inference. It should be noted that we plot the best

score obtained across all samples as objective, so even if the mean performance drops, the best sample is better owing to increase in entropy and diversity.

**Variations across initial prompts:** Another critical hyperparameter in FunSearch as outlined by Romera-Paredes et al. (2024) is the initial instance given to a FunSearch experiment. In Fig. 3d, we vary the initial program for FunSearch on the bin packing problem. We observe that a trivial instance with a more flexible structure (a for-loop adding the items $1/i$ for $i \in \{1, ..., 10\}$) starts from a low initial score but improves as more and more samples are drawn in FunSearch. Additionally, we hard-code a trivial instance as numbers without appropriate structure, and although this improves with more samples, the performance is inferior to both the trivial instance with the structure and the best known construction. Observing the output, the variations introduced by FunSearch consist of different hardcoded numbers, as opposed to inserting more structure, like loops or maths-functions, into the program. This highlights the importance of an appropriate structure and skeleton for the initial program in FunSearch. We compare this with using the best known construction (Albers et al., 2021) as the initial instance, which does start from a high score initially but stagnates quickly with iterations.

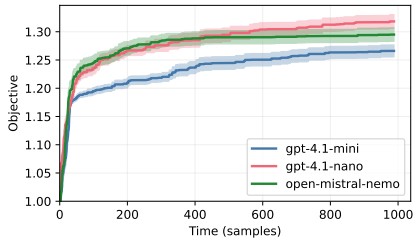

(a) Comparing different models, each with temperature 1.0 and starting with a hard-coded instance.

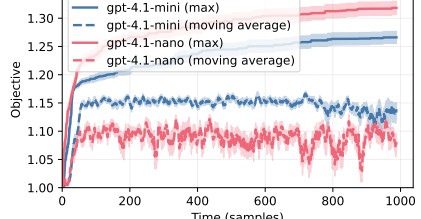

(b) Comparing rolling average (10 samples) and max-performance of gpt-4.1-mini with gpt-4.1-nano, both with temperature 1.0 and starting with a hard-coded instance.

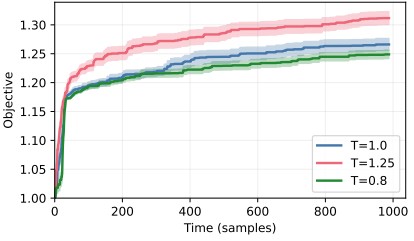

(c) Variation of different sampling temperatures for gpt-4.1-mini, each starting with a hard-coded instance.

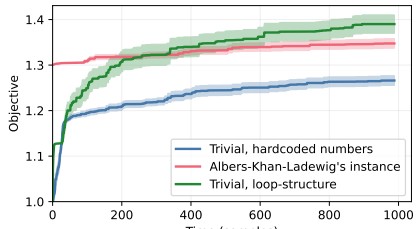

(d) Variation of initial instances for gpt-4.1-mini with temperature 1.0.

Figure 3: Comparing the effect of different hyperparameters on the objective function in bin packing.

### 3.3 CO-FUNSEARCH AND KEY RESULTS

In this section, we highlight how we used FunSearch to find instances and generalized them to achieve improved lower bounds for each problem. Furthermore, we also provide proofs for lower bounds for most of these instances.

#### 3.3.1 KNAPSACK PROBLEM

We consider the knapsack problem (as described in Section 2.2.1) as a bi-criteria optimization problem, where we want to minimize the total weight while maximizing the total profit. We used FunSearch to find instances $I$ that have a high score $\max_{1 \le i \le n} |\mathcal{P}_i(I)|/|\mathcal{P}(I)|$, i.e. where the Pareto set $\mathcal{P}_i(I)$ of a sub-instance $I_i$, which consists only of the first $i$ items of $I$, is much larger than the Pareto set $\mathcal{P}(I)$ of the entire instance $I$. The sizes of the intermittent and final Pareto-sets are obtained as a by-product of running the Nemhauser-Ullmann algorithm on $I$. Items are written as tuples of the form (weight, profit).

```python
def get_instance() -> list[tuple[int, int]]:
    """Return an instance, specified by the
    ↪  list of (weight, profit) pairs.

    Weights and profits must be non-negative
    ↪  integers.
    """
    return [(1, 2)] * 2 + [(4, 4), (2, 2),
    ↪  (1, 3)]
```

(a) Initial program.

```python
def get_instance() -> list[tuple[int, int]]:
    items = []
    n = 7
    items += [(1, 1)] * n
    items += [(4, 9)] * n
    items += [(2, 5), (3, 7)]
    return items
```

(c) An intermittent step of tuning 4b by hand.

```python
def get_instance() -> list[tuple[int, int]]:
    """Create a variant with more diverse item types and weights to potentially influence
    ↪  Pareto set size."""
    items = []
    # Repeated very light, low profit items
    items += [(1, 1)] * 8
    # Mix of moderate weight and profit items with some unique entries
    items += [(4, 9), (4, 9), (5, 10)]
    # High-profit, lightweight items with more profit variation
    items += [(2, 16), (2, 14), (3, 15)]
    # Heavier items with varied weights and higher profits to increase trade-offs
    items += [(9, 20), (12, 30), (15, 40)]
    # Small, low to moderate profit items
    items += [(1, 3), (2, 5), (3, 7), (3, 8)]
    # Very heavy, high-profit rare items with similar weights
    items += [(20, 35), (21, 36), (22, 38)]
    # Larger weight, moderate profit item to diversify options
    items += [(18, 28)]
    # Additional medium-weight high-profit items to increase complexity
    items += [(10, 25), (11, 27)]
    return items
```

(b) A program found by FunSearch after 10 trials of 500 samples each.

Figure 4: The evolution of programs generating instances of the knapsack problem. The model used was gpt-4.1-nano with a temperature of 1.0, and results obtainable despite a bug in the implementation that underestimated the sizes of some Pareto-sets.

We obtain the code in Figure 4b after running FunSearch for 10 trials of 500 samples each. Having simplified the output (shown in Fig. 4c), we can scale all items' weights up by a factor of 2 (which does not affect Pareto-optimality), decrease some profits by 1, and change the last item to obtain the following tidier instance, which achieves slightly higher scores for the same $n$:

$$\left[ \underbrace{\binom{8}{8}, ..., \binom{8}{8}}_{n \text{ times}}, \ \underbrace{\binom{2}{1}, ..., \binom{2}{1}}_{n \text{ times}}, \ \binom{4}{4}, \binom{2}{2} \right].$$

From here, we attempted to prove results about the instance. After a first draft, we found it more natural to replace the first $n$ items by $n$ powers of 2, and saw that stronger results are possible by replacing the last two items by $k$ powers of 2:

$$\left[ \binom{2^{2k}}{2^{2k}}, \binom{2^{2k+1}}{2^{2k+1}}, ..., \binom{2^{2k+n}}{2^{2k+n}}, \ \underbrace{\binom{2^k}{2^k - 1}, ..., \binom{2^k}{2^k - 1}}_{n \text{ times}}, \ \binom{2^{2k-1}}{2^{2k-1}}, \binom{2^{2k-2}}{2^{2k-2}}, ..., \binom{2^{k+1}}{2^{k+1}} \right].$$

Finally, to apply our result not only to the size of the Pareto sets but also to the runtime of the Nemhauser-Ullmann algorithm[2], we appended the factors $x_i := (1 + \frac{2^{-i}}{2^k - 1})$ to the $n$ center items:

$$\left[ \binom{2^{2k}}{2^{2k}}, ..., \binom{2^{2k+n}}{2^{2k+n}}, \ \binom{x_1 \cdot 2^k}{x_1 \cdot (2^k - 1)}, ..., \binom{x_n \cdot 2^k}{x_n \cdot (2^k - 1)}, \ \binom{2^{2k-1}}{2^{2k-1}}, ..., \binom{2^{k+1}}{2^{k+1}} \right]. \quad (1)$$

By choosing $k = \log_2(\sqrt{n}) + 1$, this instance shows:

**Theorem 3.1.** *The Nemhauser-Ullmann algorithm does not have output-polynomial running time.*

Before this work, no such instances were known. We refer to Appendix 5.1.1 for further details.

---

[2]The difference between the size of the Pareto set and the running time of the Nemhauser-Ullmann algorithm is that, for the Nemhauser-Ullmann algorithm, multiple Pareto-optimal solutions with exactly the same profit and weight are treated as a single solution for the running time.

### 3.3.2 BIN-PACKING

As outlined in Section 2.2.2, we study the Best-Fit heuristic for the bin packing problem in the random-order setting. To evaluate a generated instance, we compute the value $v_{\text{opt}}$ of an optimum solution with a solver described and implemented in Fontan & Libralesso (2020), then compute the mean $v_{\text{appx}}$ of 10,000 trials of the Best-Fit algorithm when the instance is shuffled randomly, and assign the instance a score of $\frac{v_{\text{opt}}}{v_{\text{appx}}}$. Fig. 2 shows the programs generated by FunSearch. It is straightforward to observe that Fig. 2b has multiple repetitions. We simplified this code to a list with only two parameters (Fig. 2c).

**Instance Generated by Co-FunSearch:** For fixed $m \in \mathbb{N}$, consider the instance:

$$[\underbrace{m+1, \ldots, m+1}_{m \text{ times}}, \underbrace{m, \ldots, m}_{m+1 \text{ times}}], \qquad \text{maximum bin capacity } c := m \cdot (m+1).$$

An optimal packing puts the first $m$ items into one bin, and the remaining $m + 1$ items into a second bin. This fills both bins exactly to their maximum capacity. Because $m$ and $m + 1$ are coprime, these are the only two ways of filling a bin exactly to its maximum capacity $c$. Hence, if any bin contains both an item $m$ and an item $m + 1$, the packing must use at least 3 bins. Because the instance is shuffled, Best Fit will put both an item of size $m$ and an item of size $m + 1$ into the same bin with high probability, approaching probability 1 for growing $m$. Thus, with high probability, Best-Fit will use at least 3 bins, which shows that the absolute random-order ratio of Best-Fit is at least $3/2$ (the previous best known lower bound was 1.3, by Albers et al. (2021)). Combining with the results of Dósa & Sgall (2014), we obtain the following theorem:

**Theorem 3.2.** *The absolute random-order ratio of Best-Fit is between $1.5$ and $1.7$.*

### 3.3.3 HIERARCHICAL CLUSTERING

The exact formulation of hierarchical clustering is described in Section 2.2.3. The objective for our method is to generate a clustering instance with large price of hierarchy for the $k$-median objective. To compute optimal clusterings and optimal hierarchical clusterings, we wrote a custom implementation based on Branch and Bound. The instance generated by FunSearch after 10 trials of 2,200 samples each can be seen in Fig. 5b, the trimmed instance is depicted in Fig. 5c. We generalized this to $d$ dimensions as follows:

**Instance generated by Co-FunSearch:** Let $e_i$ be the $i$th $d$-dimensional standard basis vector. For a constant $c$, consider the following weighted instance of $d + 2$ points:

$$(1, \ldots, 1), \quad (0, \ldots, 0), \quad -ce_1, \ldots, -ce_d,$$

where the point $(1, \ldots, 1)$ has weight $\infty$ and all other points have weight 1. It can be shown that:

**Theorem 3.3.** *For an appropriate choice of $c$ and for $d \to \infty$, this instance's price of hierarchy is at least $\frac{1+\sqrt{5}}{2}$, the golden ratio.*

No previous non-trivial lower bound was known for this problem. We refer the reader to Appendix 5.1.2 for further details and the proof.

### 3.3.4 GASOLINE

The details and notation of the gasoline problem are described in Section 2.2.4. We initialize the FunSearch algorithm with the instance constructed by Lorieau (2024) embedded into two dimensions as shown in Fig. 6a. Generated instances were scored by the ratio between the optimum value (computed via Gurobi Optimization, LLC (2024)) and the value returned by the iterative rounding algorithm.

**Instances generated by FunSearch:** Unlike previous problems, in this case, nearly no hand tuning was needed to modify the optimal instance generated by FunSearch. Refer to Fig. 6b and Section 5.1.3 for details.

**Theorem 3.4.** *There exist instances in every dimension $d \geq 2$ where the iterative rounding algorithm for the $d$-dimensional gasoline problem has an approximation ratio greater than 2.*

Table 2 in Appendix 5.1.3 contains the computed approximation-factors (worse than 2) for different choices of parameters. For higher $d$ and larger instances, the required computation-time quickly becomes prohibitive.

## 4 CONCLUSION AND LIMITATIONS

In this work, we use large language models (LLMs) to generate adversarial examples for heuristics addressing several well-known combinatorial optimization problems. Our approach uses FunSearch from Google Deep-Mind, a method that produces executable Python code designed to generate such adversarial instances, with

a preference for concise and interpretable implementations. This makes it relatively straightforward to understand the underlying strategies employed by the model and, in many cases, to manually generalize or refine its solutions. Traditional heuristics like local search do not converge towards such structured solutions, and understanding or generalizing their solutions is usually not feasible.

Across many of the problems we investigated, this form of human-AI collaboration enabled improvements over the existing state-of-the-art. We believe this approach is very flexible and should be considered a valuable addition to the algorithm designer's toolkit for many problems.

**Limitations.** Although our method is broadly applicable, it does not always yield improvements over the state of the art. In particular, Co-FunSearch did not produce generalizable results, or even replicate known lower bounds—for certain heuristics. These included $k$-means clustering, linkage clustering, page replacement algorithms, and the *asymptotic* random-order-ratio of best-fit. Further details are provided in the Appendix 5.4.

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

# 5 APPENDIX

## 5.1 ADDITIONAL DETAILS ON THE PROBLEMS

In this section, we discuss additional details and prove the key results.

### 5.1.1 KNAPSACK PROBLEM

In the knapsack problem, we are considering a bicriteria optimization problem, where we want to minimize the total weight while maximizing the total profit. Specifically, we are given an instance as a list of tuples of the form $(\text{weight}, \text{profit})$ from which we select a sub-list. The total weight $\text{Weight}(A)$ (respectively total profit $\text{Profit}(A)$) of a sub-list $A$ is the sum of the weights (respectively profits) of its items.

A sub-list $A$ *dominates* a sub-list $B$ if $\text{Weight}(A) \leq \text{Weight}(B)$ and $\text{Profit}(A) \geq \text{Profit}(B)$ and at least one of these inequalities is strict. A sub-list is *Pareto-optimal* if it is not dominated by any other sub-list. The *Pareto-set* $P(I)$ of an instance $I$ is the set of Pareto-optimal sub-lists of $I$. When the Pareto-set is known, objectives like the 0-1 knapsack problem "Maximise total profit while staying below a given maximum total weight $W$" can be optimised by simply finding the sub-list in $P(I)$ that has the largest total profit and whose total weight is below $W$.

As described in section 3.3.1, we obtained instance 1 via Co-FunSearch. To analyze the sizes of the instance's and subinstances' Pareto-sets, we define the two segments of the instance: For $a, b, d, n \in \mathbb{Z}_{\geq 1}$ with $d < a \leq b$, define $x_i := (1 + \frac{2^{-i}}{2^d - 1})$, and two lists:

$$I_{a,b} := \left[ \binom{2^a}{2^a}, \binom{2^{a+1}}{2^{a+1}}, \ldots, \binom{2^b}{2^b} \right], \qquad J_{d,n} := \left[ \binom{x_1 \cdot 2^d}{x_1 \cdot (2^d - 1)}, \ldots, \binom{x_n \cdot 2^d}{x_n \cdot (2^d - 1)} \right].$$

**Lemma 5.1.** *If a Pareto-optimal packing $A \in P([I_{a,b}, J_{d,n}])$ does not contain all items from $I_{a,b}$, it contains fewer than $2^{a-d}$ items from $J_{d,n}$.*

*Proof.* Subsets of $I_{a,b}$ can be represented by binary numbers of $(b - a + 1)$ bits. If $A$ does not contain all items from $I_{a,b}$ and contains at least $2^{a-d}$ items from $J_{d,n}$, we define a new packing $A'$ as follows: Increment the binary number representing $A \cap I_{a,b}$ by 1, and remove $2^{a-d}$ items from $A \cap J_{d,n}$. This changes the weights and profits by:

$$\text{Weight}(A') - \text{Weight}(A) \leq 2^a - 2^{a-d} \cdot \underbrace{\left(1 + \frac{2^{-n}}{2^d - 1}\right)}_{>1} \cdot 2^d < 0$$

$$\text{Profit}(A') - \text{Profit}(A) \geq 2^a - 2^{a-d} \cdot \left(1 + \frac{2^{-1}}{2^d - 1}\right)(2^d - 1)$$

$$= 2^a - 2^{a-d} \cdot \left(2^d - 2^{-1}\right) = 2^{a-d-1} > 0$$

Thus, $A'$ dominates $A$, and $A \notin P([I_{a,c}, J_{d,n}])$. $\qquad \square$

On the other hand, all other packings are Pareto-optimal:

**Lemma 5.2.** *If a packing $A$ of $[I_{a,b}, J_{d,n}]$ contains all items from $I_{a,b}$ or contains fewer than $2^{a-d}$ items from $J_{d,n}$, then $A$ is Pareto-optimal.*

*Proof.* All items from $I_{a,b}$ have a profit-per-weight ratio of 1, while all items from $J_{d,n}$ have a profit-per-weight ratio of $\frac{2^d - 1}{2^d} < 1$. Hence, a packing $B$ that dominates $A$ must satisfy

$$\text{Weight}(A \cap I_{a,b}) < \text{Weight}(B \cap I_{a,b}),$$

otherwise $B$ can not have enough profit to dominate $A$. If $A$ already contains all items from $I_{a,b}$, this is not possible, so only the case that $A$ contains fewer than $2^{a-d}$ items from $J_{d,n}$ remains. Due to the definition of $I_{a,b}$, the above inequality implies:

$$\text{Weight}(A \cap I_{a,b}) + 2^a \leq \text{Weight}(B \cap I_{a,b}).$$

If $B$ dominates $A$, it must hold that:

$$\text{Weight}(A \cap I_{a,b}) + \text{Weight}(A \cap J_{d,n}) \geq \text{Weight}(B \cap I_{a,b}) + \text{Weight}(B \cap J_{d,n})$$
$$\implies \text{Weight}(A \cap J_{d,n}) - 2^a \geq \text{Weight}(B \cap J_{d,n}).$$

But $A$ contains fewer than $2^{a-d}$ items from $J_{d,n}$, so:

$$\text{Weight}(A \cap J_{d,n}) \leq 2^{a-d} \cdot \left(1 + \frac{2^{-1}}{2^d - 1}\right) \cdot (2^d - 1) = 2^a - 2^{a-d-1} < 2^a.$$

This implies $0 > \text{Weight}(B \cap J_{d,n})$, a contradiction. $\qquad\square$

Hence, we can describe the Pareto-set exactly:

$$P([I_{a,b}, J_{d,n}]) = \{A \cup B \mid A \subsetneq I_{a,b},\ B \subseteq J_{d,n},\ |B| < 2^{a-d}\} \ \dot{\cup}\ \{I_{a,b} \cup B \mid B \subseteq J_{d,n}\}.$$

Its size is (using notation involving binomial coefficients, not vectors):

$$|P([I_{a,b}, J_{d,n}])| = (2^{b-a+1} - 1) \cdot \left\lceil \sum_{i=0}^{\min(n, 2^{a-d}-1)} \binom{n}{i} \right\rceil + 2^n.$$

For $k, n \in \mathbb{N}$ with $2^k \leq n/2$, consider two instances:

$$\mathbb{I}_1 := [I_{2k,\, 2k+n},\ J_{k,n}],$$

$$\mathbb{I}_2 := \left[\mathbb{I}_1,\ \binom{2^{k+1}}{2^{k+1}}, \binom{2^{k+2}}{2^{k+2}}, \ldots, \binom{2^{2k-1}}{2^{2k-1}}\right].$$

$\mathbb{I}_1$ is a sub-instance of $\mathbb{I}_2$. $\mathbb{I}_2$ (which is exactly instance 1) contains the same items as $[I_{k+1,\, 2k+n},\ J_{k,n}]$. The sizes of their Pareto-sets can be bounded by:

$$|P(\mathbb{I}_1)| \geq (2^{n+1} - 1) \cdot \binom{n}{2^k - 1} + 2^n \geq (2^{n+1} - 1) \cdot \left(\frac{n}{2^k - 1}\right)^{(2^k-1)}.$$

$$|P(\mathbb{I}_2)| \leq (2^{k+n} - 1) \cdot (n + 1) + 2^n \leq (2^{k+n} - 1) \cdot (n + 2)$$

The ratio between the two sizes is:

$$\frac{|P(\mathbb{I}_1)|}{|P(\mathbb{I}_2)|} \geq \frac{2^{n+1} - 1}{2^{k+n} - 1} \cdot \left(\frac{n}{2^k - 1}\right)^{(2^k-1)} \cdot \frac{1}{n + 2}$$

For $k = \log_2(\sqrt{n}) + 1$, we obtain:

$$\frac{|P(\mathbb{I}_1)|}{|P(\mathbb{I}_2)|} \geq \frac{2^{n+1} - 1}{(\sqrt{n} + 1) \cdot 2^n - 1} \cdot \left(\frac{n}{\sqrt{n}}\right)^{\sqrt{n}} \cdot \frac{1}{n + 2} = \Theta(n^{(\sqrt{n}-3)/2}).$$

The length of the instance $\mathbb{I}_2$ is not $n$ but $m := |\mathbb{I}_2| = 2n + k$, resulting in a lower bound of $O\left(\left(\frac{m}{2}\right)^{(\sqrt{m/2}-3)/2}\right)$.

In implementations of the Nemhauser-Ullmann algorithm, two Pareto-optimal packings can be treated as equivalent if they have the same total weight and total profit. Hence, the runtime can be upper-bounded not only by the sum of the sizes of the Pareto-sets $|P(I_{1:1})| + \ldots + |P(I_{1:n})|$, but even the sizes of the Pareto-sets when two packings with the same total weight and total profit are treated as identical. The only purpose of the leading factors $\left(1 + \frac{2^{-n}}{2^d - 1}\right)$ in $J_{d,n}$ is to prevent two Pareto-optimal packings from having the same total profit. As a consequence, we also obtain a bound of $O\left(\left(\frac{m}{2}\right)^{(\sqrt{m/2}-3)/2}\right)$ for the runtime of the Nemhauser-Ullmann algorithm.

**Lemma 5.3.** *If $A, B \subseteq [I_{a,b}, J_{d,n}]$ are two distinct Pareto optimal packings, then* $\text{Profit}(A) \neq \text{Profit}(B)$.

*Proof.* Because both $A$ and $B$ are Pareto-optimal, we know by 5.1 that $|A \cap J_{d,n}| < 2^{a-d}$ (same for $B$), hence:

$$\text{Profit}(A \cap J_{d,n}) < 2^{a-d} \cdot \left(1 + \frac{2^{-1}}{2^d - 1}\right) \cdot (2^d - 1)$$

$$= 2^{a-d} \cdot \left(2^d - \frac{1}{2}\right)$$

$$= 2^a - 2^{a-d-1} < 2^a.$$

(same for $\text{Profit}(B \cap J_{d,n})$).

- If $A \cap I_{a,b} \neq B \cap I_{a,b}$, the difference between $\text{Profit}(A \cap I_{a,b})$ and $\text{Profit}(B \cap I_{a,b})$ would be at least $2^a$, due to the definition of $I_{a,b}$. In this case, the above inequality already shows $\text{Profit}(A) \neq \text{Profit}(B)$.
- If $A \cap I_{a,b} = B \cap I_{a,b}$, then $A \cap J_{d,n} \neq B \cap J_{d,n}$, and we need to show that $\text{Profit}(A \cap J_{d,n}) \neq \text{Profit}(B \cap J_{d,n})$. This is equivalent to showing that any two distinct subsets of:

$$\{(2^d - 1) + 2^{-1},\ (2^d - 1) + 2^{-2},\ \ldots,\ (2^d - 1) + 2^{-n}\},$$

have a distinct sum. This holds because the total sum of the summands $2^{-1}, \ldots, 2^{-n}$ is always smaller than 1, whereas $2^d - 1 \geq 1$.

$\qquad\square$

## 5.1.2 HIERARCHICAL CLUSTERING

In clustering, we're given a set of $n$ weighted points and a number $k$, with the task of finding a partition of the set of points into $k$ *clusters* such that the total cost of the clusters is small. In $k$-median clustering, the points are a finite subset of $\mathbb{R}^d$ and the cost of a cluster $C$ is defined as the sum of the weighted $L^1$-distances all points have to the center, where the center is the best possible choice within that cluster:

$$\text{Cost}(C) = \min_{p \in C} \sum_{q \in C} w(q)\|p - q\|_1$$

Here, $w(q)$ is the weight of $q$ as specified by the instance. The total cost of a clustering is the sum of the costs of its clusters.

Clustering is used to analyze empirical data, but it's usually not clear what number of clusters $k$ is a good choice for the dataset. Instead of computing a clustering for a fixed $k$, one could instead compute a *Hierarchical Clustering*, which has a clustering for each $k \in \{1, ..., n\}$ and these clusterings are nested: A hierarchical clustering $H = (H_1, ..., H_n)$ consists of $n$ clusterings such that, for all $i \in \{2, ..., n\}$, $H_i$ is obtained by merging two clusters of $H_{i+1}$.

While hierarchical clusterings have an intuitive structure and don't require us to decide on a number $k$ of clusters beforehand, they come at the disadvantage of their clusters $H_i$ possibly having a higher cost than the *optimal* $i$-clustering, because optimal clusterings need not form a hierarchy. For a given instance (a finite set of points in $\mathbb{R}^d$) $I$, we can measure the performance of a hierarchical clustering $H$ by comparing each of its clusterings $H_i$ to the best $i$-clustering, and choosing the level where this ratio is highest.

To measure how much we sacrifice when restricting ourselves to hierarchical clusterings on an instance $I$, we consider the *Price of Hierarchy of $I$* as the best hierarchical clustering according to that measure:

$$\text{PoH}(I) := \min_H \max_{k \in \{1,...,n\}} \left[ \frac{\text{Cost}(H_k)}{\text{Cost}(\text{OPT}_k)} \right],$$

where $\text{OPT}_k$ denotes an optimal $k$-clustering for $I$.

The *Price of Hierarchy for $k$-median clustering* $\text{PoH}_{k\text{-median}}$ denotes the worst-case Price of Hierarchy of $I$ across all possible instances $I$. Thus, $\text{PoH}_{k\text{-median}}$ captures the worst-case quality of an optimal hierarchical clustering when compared to an optimal non-hierarchical clustering.

Fix the dimension $d \geq 4$. Put $c := \frac{\sqrt{4d^2+(3-d)^2}+d-3}{2}$, which is one of the two roots of $0 = c^2 - c(d-3) - d^2$. Because $d \geq 4$, we know that $5d^2 - 6d \geq 4d^2$, hence:

$$c = \frac{\sqrt{4d^2 + (d-3)^2} + d - 3}{2} > \frac{2d + d - 3}{2} > d.$$

Let $e_i$ be the $i$th $d$-dimensional standard basis vector. Consider the following weighted instance of $d+2$ points:

$$(1, \ldots, 1), \quad (0, \ldots, 0), \quad -ce_1, \ldots, -ce_d,$$

where the point $(1, \ldots, 1)$ has weight $\infty$ and all other points have weight 1.

**Theorem 5.4.** *For $k$-median clustering, this instance's price of hierarchy is at least $\frac{c}{d}$.*

*Proof.* For contradiction, assume there exists a hierarchical clustering $H = (H_1, \ldots, H_{d+2})$ such that, on every level, the cost of $H_k$ is strictly less than $\frac{c}{d}$ times the cost of the best clustering using $k$ clusters. This enables us to narrow down the structure of $H$:

- For $k = d + 1$, there is one cluster $C$ containing two points, while all other clusters contain only a single point. Depending on which two points constitute $C$, we can calculate the total cost of the clustering:
  - If $C = \{(0, \ldots, 0), (1, \ldots, 1)\}$, the total cost is:
    $$\|(0, \ldots, 0) - (1, \ldots, 1)\|_1 = d.$$
    - If $C = \{(0, \ldots, 0), -ce_i\}$ for some $i$, the total cost is $c$.
    - If $C = \{(1, \ldots, 1), -ce_i\}$ for some $i$, the total cost is $d + c$.
    - If $C = \{-ce_i, -ce_j\}$ for some $i \neq j$, the total cost is $2c$.
  - Because $d < c$, this constrains $H_k$ to $C = \{(0, \ldots, 0), (1, \ldots, 1)\}$, otherwise the total cost of $H_k$ would be at least $\frac{c}{d}$ times the cost of an optimal $(d + 1)$-clustering.
- For $k = 2$: The clustering now contains exactly two clusters. Because $H$ is a hierarchical clustering, we now know that $H_2$ has a cluster that contains $(0, \ldots, 0)$, $(1, \ldots, 1)$ and some number $0 \leq n \leq d - 1$ of the $-ce_i$, while its other cluster contains the remaining $d - 1 - n$ of the $-ce_i$. Due to

symmetry, this number $n$ is sufficient for calculating the total cost of $H_2$. Because $(1, \ldots, 1)$ has infinite weight, this point must be the center of the first cluster, so this cluster has cost:

$$\big\|(1, \ldots, 1) - (0, \ldots, 0)\big\|_1 + n \cdot \big\|(1, \ldots, 1) - (-ce_1)\big\|_1 = d + n \cdot (c + d)$$

The cluster containing the remaining $d - 1 - n$ of the $-ce_i$ can choose any point as its center. It has cost:

$$(d - 2 - n) \cdot \big\|ce_1 - ce_2\big\|_1 = (d - 2 - n) \cdot 2c$$

Given $n$, the total cost of $H_2$ is $d + c(2d - 4) + n(d - c)$. Because $d - c < 0$, the best choice for $n$ would be $n = d - 1$, resulting in a cost of $c(d - 3) + d^2$. This is only a lower bound on the cost of $H_2$, because other levels in the hierarchy might put additional constraints on $H_2$.

For an *upper* bound on the *optimal* cost of a 2-clustering, consider the clustering that has $(1, \ldots, 1)$ in its first cluster, and all other points in its second cluster. By assuming the center of the second cluster is $(0, \ldots, 0)$, we get an upper bound on the total cost of this clustering of:

$$d \cdot \big\|(0, \ldots, 0) - (-ce_1)\big\|_1 = d \cdot c.$$

Hence, the ratio between the cost of $H_2$ and the cost of an optimal 2-clustering is at least:

$$\frac{c(d - 3) + d^2}{d \cdot c} = \frac{d - 3}{d} + \frac{d}{c}$$

We defined $c$ as one of the roots of $0 = c^2 - c(d - 3) - d^2$. Dividing out $cd$, we get $\frac{d-3}{d} + \frac{d}{c} = \frac{c}{d}$. However, this contradicts the assumption that the ratio between $H_2$ and an optimal 2-clustering is strictly less than $\frac{c}{d}$.

Thus, the instance's price of hierarchy is at least $\frac{c}{d}$. $\qquad\square$

For large $d$, this fraction $\frac{c}{d} = \frac{\sqrt{4d^2 + (3-d)^2} + d - 3}{2d}$ converges to $\frac{1 + \sqrt{5}}{2}$, the golden ratio.

### 5.1.3 GASOLINE

In the generalised Gasoline problem, we are given two sequences of $d$-dimensional vectors $X = (x_1, ..., x_n) \in \mathbb{N}_{\geq 0}^{d \times n}$ and $Y = (y_1, ..., y_n) \in \mathbb{N}_{\geq 0}^{d \times n}$ which sum to the same total: $\sum_{i=1}^{n} x_i = \sum_{i=1}^{n} y_i$. Our task is to find a permutation $\pi$ of the $x_i$ that minimises:

$$\min_{\pi \in S_n} \sum_{j=1}^{d} \left[ \max_{1 \leq k \leq n} \left( \sum_{i=1}^{k} x_{\pi(i)} - \sum_{i=1}^{k-1} y_i \right)_j - \min_{1 \leq k \leq n} \left( \sum_{i=1}^{k} x_{\pi(i)} - \sum_{i=1}^{k} y_i \right)_j \right]$$

This can be written as an ILP, with a permutation-matrix $Z$ as a free variable. Let $\mathbf{1}$ be the vector containing a 1 in every entry.

$$\min \|\beta - \alpha\|_1 \quad \text{s.t.}$$

$$\sum_{l=1}^{n} \sum_{i=1}^{m} x_l Z_{il} - \sum_{i=1}^{m-1} y_i \leq \beta \quad \forall m \in [n]$$

$$\sum_{l=1}^{n} \sum_{i=1}^{m} x_l Z_{il} - \sum_{i=1}^{m} y_i \geq \alpha \quad \forall m \in [n]$$

$$Z\mathbf{1} \leq \mathbf{1}$$

$$\mathbf{1}^T Z \leq \mathbf{1}^T$$

$$Z \in \{0, 1\}^{n \times n}$$

$$\alpha, \beta \in \mathbb{R}^d$$

In the $i$th step of the iterative rounding algorithm, the columns $1, ..., i - 1$ of $Z$ have already been fixed to integral values by the previous steps and, for column $i$, we attempt to insert every possible unit-vector (which does not conflict with the fixed rows and the permutation-matrix requirement) and then solve the Linear Program obtained by removing the integrality-requirements on columns $i + 1, ..., n$. We then fix column $i$ of $Z$ to that unit-vector which yielded the best value for the relaxed LP, breaking ties by preferring unit-vectors where the 1 occurs earlier. After the $n$th step of this algorithm, $Z$ is fixed entirely to a permutation-matrix.

(Rajković, 2022, Conjectures 2 and 3) conjectured that this algorithm is a 2-approximation for $d \geq 2$, which FunSearch found a counterexample for.

| $d$ | $k$ | Length of $X$ | Iterative-Rounding | Optimum | Iterative-Rounding/Optimum |
|---|---|---|---|---|---|
| 2 | 2 | 6 | 10 | 8 | 1.25 |
| 2 | 3 | 14 | 26 | 12 | 2.1667 |
| 2 | 4 | 30 | 58 | 20 | 2.9 |
| 2 | 5 | 62 | 122 | 36 | 3.389 |
| 2 | 6 | 126 | 250 | 68 | 3.676 |
| 3 | 2 | 12 | 18 | 12 | 1.5 |
| 3 | 3 | 28 | 42 | 16 | 2.625 |
| 3 | 4 | 60 | 90 | 24 | 3.75 |
| 3 | 5 | 124 | 186 | 40 | 4.65 |
| 4 | 2 | 18 | 24 | 16 | 1.5 |
| 4 | 3 | 42 | 56 | 20 | 2.8 |
| 4 | 4 | 90 | 120 | 28 | 4.286 |

Table 2: The approximation-factor of the Iterative-Rounding algorithm on the instances found by FunSearch.

Fix some $k \in \mathbb{N}$. For any $i$, define $u_i := 2^k(1 - 2^{-i})$. Let $\oplus$ denote list-concatenation. The 1-dimensional instance found by Lorieau (2024) can be written as follows:

$$X = \left( \bigoplus_{i=1}^{k-1} \bigoplus_{1}^{2^i} [u_i] \right) \oplus \left( \bigoplus_{1}^{2^k-1} [2^k] \right) \oplus [0]$$

$$Y = \bigoplus_{i=1}^{k} \bigoplus_{1}^{2^i} [u_i]$$

Let $e_j$ be the $j$th unit-vector. FunSearch extended the instance to $d$ dimensions as follows:

$$X := \left( \bigoplus_{i=1}^{k-1} \bigoplus_{1}^{2^i} \bigoplus_{j=2}^{d} [u_i e_1 + 4e_j] \right) \oplus \left( \bigoplus_{j=2}^{d} \left( \bigoplus_{1}^{2^k-1} [2^k e_1] \right) \oplus [4e_j] \right)$$

$$Y := \bigoplus_{i=1}^{k} \bigoplus_{1}^{2^i} \bigoplus_{j=2}^{d} [u_i e_1 + 2e_j]$$

Table 2 contains computed approximation-factors for different choices of $d$ and $k$. For higher $d$ and $k$, the instances quickly grow prohibitively large.

In our computational experiments, both APX and OPT scale linearly with input-length $|X|$:

$$\text{APX} = O(1) + |X| \cdot \begin{cases} 2 & d = 2 \\ 3/2 & d = 3 \ , \\ 4/3 & d = 4 \end{cases} \qquad \text{OPT} = O(1) + |X| \cdot \begin{cases} 1/2 & d = 2 \\ 1/4 & d = 3 \\ 1/6 & d = 4 \end{cases}$$

If this scaling held for larger $k$, the approximation-factors would approach $4, 6, 8$ for $d = 2, 3, 4$ respectively. Unfortunately, the proof-strategy employed in Lorieau (2024) does not apply here, as the optimum value of the relaxed Linear Program changes at each step of the algorithm. Hence, we are unable to provide a proof that these trends hold asymptotically.

## 5.2 PROGRAMS FOUND BY FUNSEARCH AND CO-FUNSEARCH

In this seciton, we outline the programs generated by FunSearch and how these were simplified by experts for hierarchical clustering and gasoline respectively in figure 5 and 6 respectively.

## 5.3 COMPARING LOCAL VS FUNSEARCH

We list the instance list generated by local-search and fun-search in table 3. It can be clearly seen here that local search instance has no discernible structure while funsearch instance has a structure which can be further improved by domain experts.

## 5.4 LIMITATIONS OF CO-FUNSEARCH

Below, we note some combinatorial optimization problems and their corresponding heuristics where our methods did not yield improvements to existing lower bounds:

```
def get_weighted_points() ->
↪ list[tuple[float, np.ndarray]]:
    """Return a new weighted
    ↪ clustering-problem, specified by a
    ↪ list of weighted points.
    The returned tuple consists of the weight
    ↪ of the point, and the point
    ↪ itself."""
    weighted_points = [(1.0, np.array([0, 0,
    ↪ 0, 0])), (1e8, np.array([1, 0, 0,
    ↪ 0]))]
    return weighted_points
```

(a) The initial program given to FunSearch.

```
def get_weighted_points() ->
↪ list[tuple[float, np.ndarray]]:
    return [
        (1.0, np.zeros(14)),
        *[(1.0, -np.eye(14)[i]) for i in
        ↪ range(14)],
        (1e10, np.ones(14) / 20),
    ]
```

(c) The result of tuning by 5b by hand.

```
def get_weighted_points() -> list[tuple[float, np.ndarray]]:
    """Return a new weighted clustering-problem, specified by a list of weighted points.
    The returned tuple consists of the weight of the point, and the point itself."""
    return [
        (1.0, np.zeros(14)),
        (1e10, np.ones(14)),
        *[(1.0, np.eye(14)[i]) for i in range(7)],
        *[(1.0, np.eye(14)[i]*-1) for i in range(7, 13)],
        *[(1e10-i*1e9, np.linspace(i*0.1, (i+1)*0.1, 14, endpoint=False)) for i in range(7)],
        (1e11, np.array([13, 12, 11, 10, 9, 8, 7, 6, 5, 4, 3, 2, 1, 0])),
        (1e12, np.array([0, 1, 2, 3, 4, 5, 6, 7, 8, 9, 10, 11, 12, 13])),
        (1e13, np.array([1, 2, 3, 4, 5, 6, 7, 8, 9, 10, 11, 12, 13, 14])*10),
        (1e14, np.array([14, 13, 12, 11, 10, 9, 8, 7, 6, 5, 4, 3, 2, 1])*100),
        (1e15, np.array([1, 1, 1, 1, 1, 1, 1, 1, 1, 1, 1, 1, 1, 1])*1000),
    ]
```

(b) A program found by FunSearch after 10 trials of 2,200 samples each.

Figure 5: The evolution of programs generating clustering-instances. The model used was open-mistral-nemo with a temperature of 1.5.

| | Local Search | FunSearch | Co-FunSearch |
|---|---|---|---|
| Items | 0.003 | 0.08 | 0.167 |
| | 0.005 | 0.08 | 0.167 |
| | 0.006 | 0.08 | 0.167 |
| | 0.007 | 0.08 | 0.167 |
| | 0.021 | 0.08 | 0.167 |
| | 0.068 | 0.114 | 0.167 |
| | 0.073 | 0.114 | 0.143 |
| | 0.170 | 0.114 | 0.143 |
| | 0.202 | 0.114 | 0.143 |
| | 0.219 | 0.114 | 0.143 |
| | 0.306 | 0.114 | 0.143 |
| | 0.375 | 0.114 | 0.143 |
| | 0.540 | 0.2 | 0.143 |
| | | 0.6 | |
| Score | 1.4938 | 1.4979 | 1.4977 |

Table 3: Comparing the final instances found by local search, FunSearch and Co-FunSearch for the randomised Best-Fit bin-packing problem.

```python
def gasoline(n: int) -> tuple[list[np.ndarray], list[np.ndarray]]:
    """Return a new gasoline-problem, specified by the two lists of
    ↪ 2d-non-negative-integer-points.
    Both lists must have length at most n and consist only of points in N^2.
    """
    k = int(math.log2(n + 2)) - 1
    xs, ys = [], []
    for i in range(1, k):
        rounded = int(2**k * (1 - 2 ** (-i)))
        xs.extend([np.array([rounded, 0]) for _ in range(2**i)])
        ys.extend([np.array([rounded, 0]) for _ in range(2**i)])

    xs.extend([np.array([2**k, 0]) for _ in range(2**k - 1)])
    xs.append(np.array([0, 0]))

    rounded = int(2**k * (1 - 2 ** (-k)))
    ys.extend([np.array([rounded, 0]) for _ in range(2**k)])

    return xs, ys
```

(a) The initial program given to FunSearch. This is the construction of Lorieau (2024) embedded into $\mathbb{R}^2$.

```python
 def gasoline(n: int) -> tuple[list[np.ndarray], list[np.ndarray]]:
     """Yet another variation of the gasoline-problem generator."""
     k = int(math.log2(n + 2)) - 1
     xs, ys = [], []
     for i in range(1, k):
         rounded = int(2**k * (1 - 2 ** (-i)))
         xs.extend([np.array([rounded, 0]) for _ in range(2**i)])
-        ys.extend([np.array([rounded, 0]) for _ in range(2**i)])
+        ys.extend([np.array([rounded, 2]) for _ in range(2**i)])  # No change

-    xs.extend([np.array([2**k, 0]) for _ in range(2**k - 1)])
+    xs.extend([np.array([2**k, 4]) for _ in range(2**k - 2)])  # No change
-    xs.append(np.array([0, 0]))
+    xs.append(np.array([0, 1]))  # Changed from [0, 2] to [0, 1]
+    xs.append(np.array([2**k, 2]))  # Changed from [2**k, 0] to [2**k, 2]

     rounded = int(2**k * (1 - 2 ** (-k)))
-    ys.extend([np.array([rounded, 0]) for _ in range(2**k)])
+    ys.extend([np.array([rounded, 2]) for _ in range(2**k - 1)])  # No change
+    ys.append(np.array([0, 1]))  # Changed from [0, 2] to [0, 1]
```

(b) The difference between the initial program and a program found by FunSearch after 10 trials of 950 samples each, which we only tuned by discarding the final element of both lists.

Figure 6: The evolution of programs generating 2-dimensional gasoline-instances. The model used was open-mistral-nemo with a temperature of $1.5$. Lists were clipped to length $n$ before evaluation.

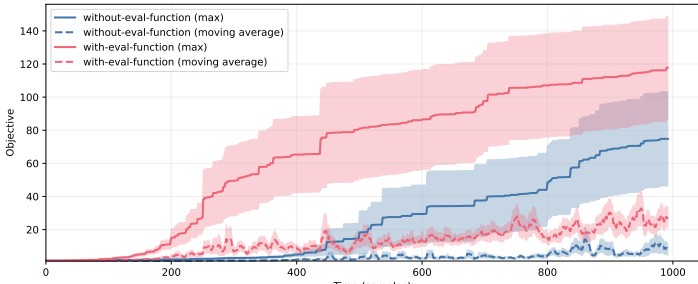

Figure 7: As mentioned previously, we did usually not include the evaluation-function in the prompt. We re-ran two FunSearch-experiments across 30 trials each, one including the evaluation-function and one not including the evaluation-function, on the Knapsack-Problem (the other evaluation-functions delegated a lot of their work to external solvers). We plot both the average running maximum across the trials, and the rolling average across 10 samples, including the standard error. Although the standard-errors are large, the trials including the evaluation-function did perform better on the objective. However, as it used more tokens when querying the LLM, the experiment including the evaluation-function was about 1.6 times as expensive as the experiment omitting the evaluation-function. All best-performing programs across both experiments had a similar structure.

- Better heuristics for page replacement algorithms (evaluated on synthetic and real data), but FunSearch consistently converged to the existing NFU heuristic.
- Lower bounds on the Price of Hierarchy of $k$-means clustering (as opposed to $k$-median clustering).
- Lower bounds on the price of Ward's method for hierarchical 2-dimensional $k$-means clustering: Instead of comparing the best possible hierarchical clustering to the optimal clusterings, we compare the hierarchical clustering found by starting with each point in a singleton cluster, and iteratively merging the pair of clusters which result in the lowest objective. Neither FunSearch nor local search managed to recover the State of the Art when starting from a trivial instance. When starting from the State of the Art in 2 dimensions, both FunSearch and local search improved it marginally (FunSearch less so than local search, even after tuning), but not in a generalisable way.
- Lower bounds on the *asymptotic* random-order-ratio of Best-Fit, which is the same as the absolute random-order-ratio but restricted to only "large" instances (Albers et al., 2021). FunSearch did not find any interpretable instances improving on the state of the art.