# OpenReview forum: "Adversarial examples for heuristics in combinatorial optimization: An LLM based approach"
_ICLR.cc/2026/Conference — ICLR 2026 Conference Withdrawn Submission_

### Official Review · Reviewer_QmxA · 2025-10-30

**Soundness:** 2
**Presentation:** 3
**Contribution:** 2
**Rating:** 2
**Confidence:** 4

**Summary:**

This paper applies FunSearch (LLM-based evolutionary program search) to generate adversarial instances for combinatorial optimization heuristics. The authors investigate four problems: Nemhauser-Ullmann knapsack heuristic, Best-Fit bin packing, k-median hierarchical clustering, and iterative rounding for the gasoline problem. They propose "Co-FunSearch" combining automated generation with manual refinement.

**Strengths:**

1. Rigorous mathematical follow-up: Unlike pure black-box approaches, the authors provide formal proofs for most claims (Theorems 3.1-3.4).

2. Honest reporting: The limitations section (5.4) honestly reports failures.

3. Diverse problem selection: Four different combinatorial optimization domains demonstrate some generality.

**Weaknesses:**

1. The paper doesn't establish why finding adversarial instances via LLM is an important research problem. The paper doesn't demonstrate that these adversarial instances have any practical implications. Are these contrived constructions or do they reveal fundamental algorithmic weaknesses?

2. Results may already be known or trivial: The bin packing instance is suspiciously simple - the authors should check whether this construction appears implicitly in prior work.

3. Gasoline extrapolation from Table 2 to general claims lacks justification. Need either formal proof of scaling or results for larger instances.

4. "Co-" prefix admits heavy expert refinement needed, contradicting automated discovery narrative. Contribution breakdown between LLM and human unclear.

**Questions:**

Given that (1) heavy human expert refinement is required ("Co-"), (2) success rate is ~50% or lower, (3) several results have questionable novelty/correctness, and (4) practical impact is unclear - can you articulate a clear value proposition for why the community should pursue LLM-based adversarial instance generation over traditional mathematical analysis? What specific advantages does this approach offer that justify its costs and limitations?

---

> ### Author Response · Authors · 2025-11-21
>
> **Query**:  The paper doesn't establish why finding adversarial instances via LLM is an important research problem. The paper doesn't demonstrate that these adversarial instances have any practical implications. Are these contrived constructions or do they reveal fundamental algorithmic weaknesses?
>
> **Response**: Finding lower bounds for algorithms is an integral part of algorithmic research and usually goes hand in hand with developing better algorithms. For example, when developing new approximation algorithms for a problem, one is often faced with the situation that one has developed an algorithm that one suspects to achieve a certain approximation factor. Then the typical approach of an algorithms' researcher is to try simultaneously to prove an upper bound and to find a lower bound. While arguably, finding an upper bound is usually the more pleasant outcome, also lower bounds give valuable insights because they show that the algorithm needs to be adapted to achieve a better upper bound and sometimes they even show how the algorithm needs to be modified. The same reasoning applies also to the analysis of the running time of an algorithm instead of the approximation factor.
>
> One of very many prominent prototypical examples where researchers have devoted a significant amount of time on finding lower bounds for algorithms is the Greedy algorithm for the Shortest Common Superstring Problem: https://arxiv.org/abs/2407.20422 Other important examples include lower bounds for the running time of the policy iteration algorithm for MDPs (https://arxiv.org/abs/2506.12254) and lower bounds for various pivot rules for the simplex algorithm (https://link.springer.com/chapter/10.1007/978-3-031-93112-3_7).
>
> We adopt in our submission the classical worst-case perspective from theoretical computer science. Indeed this implies that lower bound examples are sometimes contrived and do not always reflect typical inputs. This problem is addressed by other models like smoothed analysis but worst-case analysis is still the dominant measure in the analysis of algorithms. Our work is meaningful and highlights fundamental algorithmic weaknesses in existing heuristics. Adapting it to other models like smoothed analysis is an interesting topic for future research.
>
> **Query**: Results may already be known or trivial: The bin packing instance is suspiciously simple - the authors should check whether this construction appears implicitly in prior work.
>
> **Response**: This point precisely highlights the power of collaborating with FunSearch. We were ourselves surprised at the simplicity of the tight lower bound but recent work (Best fit bin packing with random order revisited. Algorithmica, 2021 by Albers et al.) did not discover these lower bounds. We rechecked the literature and could not find any instance, we will be happy to incorporate a reference of such an instance in the existing literature.
>
> **Query**: Gasoline extrapolation from Table 2 to general claims lacks justification. Need either formal proof of scaling or results for larger instances.
>
> **Response**:
> We changed the remark in the appendix to clarify that the extrapolation is mere conjecture, and that we have been unable to prove it.
>
>
> **Query**: "Co-" prefix admits heavy expert refinement needed, contradicting automated discovery narrative. Contribution breakdown between LLM and human unclear.
>
> **Response**: We haven’t claimed fully automated discovery anywhere in the paper. At each step, we have tried to be very clear that this is a collaboration between the automated discovery of good examples with human interaction used for generalizing them to our final result. The function of FunSearch is the generation of structured instances that are then analyzed and generalized by a human researcher. We mention this in our abstract (lines 20-23), and then repeat this in our explanation of the methodology (lines 75-85 in modified paper or lines 68-78 in original submission), where we explain that both automated discovery using FunSearch as well as human analysis are essential for our approach. We emphasize that for each of the four problems highlighted in our work, the state-of-the-art was improved using this methodology.

---

> > ### Author Response · Authors · 2025-11-21
> >
> > **Query**: Given that (1) heavy human expert refinement is required ("Co-"), (2) success rate is ~50% or lower, (3) several results have questionable novelty/correctness, and (4) practical impact is unclear - can you articulate a clear value proposition for why the community should pursue LLM-based adversarial instance generation over traditional mathematical analysis? What specific advantages does this
> >
> > **Response**: (1) While we agree that human expert refinement is an important component of the Co-FunSearch framework, this is a common trend in obtaining state-of-the-art in recent works: (a) Terence Tao’s latest publication on Nikodym set constructions (https://arxiv.org/abs/2511.07721) (b) Advancing mathematics by guiding human intuition with AI (https://www.nature.com/articles/s41586-021-04086-x). Current AI systems lack the capacity to autonomously resolve complex open problems in theoretical computer science and mathematics. These papers and ours suggest a collaboration between experts and AI offer complementary advantages for solving such challenges.
> >
> > (2) The success rate is low, but the methodology is general enough to be tried on many other problems. We would argue that making progress on even one or two of these problems is quite impressive since many of the results we improve upon have been open problems in theoretical computer science. Our analysis covers the complete range of results on these challenges, revealing the true strengths and weaknesses of current systems.
> >
> > (3) We provide proofs of correctness for each of the results and improve upon the existing state-of-the-art, many of which have been published at top tier conferences in theoretical computer science. We failed to understand why our contributions lack novelty and correctness and disagree with the reviewer on this point.
> >
> > (4) As described in (1) and (2), both LLMs and human experts are necessary for improvements to SOTA on these problems using our methodology. We do not claim that standalone LLM based approaches are preferable over mathematical analysis, rather a collaboration could achieve much superior results. It is not clear why our approach seems expensive to the reviewer. We would also like to highlight that the expense of LLM API calls is decreasing rapidly.

---

> > > ### Comment · Reviewer_QmxA · 2025-11-26
> > >
> > > I thank the authors for their detailed response. I will keep my score.
> > >
> > > 1. Expert Dependency: "Co-FunSearch" still heavily relies on human experts to interpret and generalize LLM-generated instances. If the core theoretical insights are primarily attributed to humans, the technical contribution of the automated framework itself appears weak.
> > >
> > > 2. Depth & Generalization: While the Bin Packing counterexample is new, its structure is overly simple. Regarding the Gasoline problem, the extrapolation is admittedly a "conjecture" without proof. Furthermore, the intrinsic difficulty of many combinatorial problems suggests the method may not be as effectively transferable to other scenarios as claimed.
> > >
> > > 3. Overall Assessment: Aligning with other reviewers regarding limited technical novelty, the paper presents more of an LLM-assisted case study rather than a fully formed, deep automated methodology.

---

### Official Review · Reviewer_cqeS · 2025-10-30

**Soundness:** 2
**Presentation:** 2
**Contribution:** 3
**Rating:** 2
**Confidence:** 2

**Summary:**

This paper considers the problem of finding hard instances for a fixed algorithm for a fixed problem. The studied problems are all NP-hard, namely Knapsack, Bin packing, k-median in hierarchical clustering, and Lovasz's gazoline problem. Specific approximation algorithms are considered for each of these problems. The goal is to use artificial intelligence to generate instances which maximize the approximation ratio of the considered algorithms. Instead of generating the instance itself, a compact description is generated using a large language model artificial intelligence. The prompt asks the AI to come up with a worse instance, and provides a first python program generating an instance.

I find this approach quite crazy, in my own humble experience, lower bound instances have to be constructed by hand, and computers turned out to be bad in generating hard instances. But investigating the power of LLMs for this task is an interesting approach.

**Strengths:**

The paper succeeds to find an instance which breaks a conjecture from 2024 about a specific algorithm for the gazoline problem.  For other problems the LLM gave the authors ideas to improve the generated instance. So this work shows that LLM could help in algorithm design. I don't know much about LLMs, so I don't have the right background to evaluate the paper. But I am into algorithm design, and hence any new method that could help is welcomed.

**Weaknesses:**

Overall I am still hungry after reading the paper. From what I have seen in the algorithm community, is that most hard instances consists of some fractions, to which some epsilon values have been added or removed. Also I think the approach could benefit from the design of a grammar describing compactly instances, rather than generating a Python program, because the search space is more targeted to a form of aimed instances.

**Questions:**

You don't give the evaluation function to the LLM. Explain this choice.

Line 372, explain what each dimension means. I guess first is weight, and second is profit.
Several times you mention a choice of temperature, which I only know in the context of simulated annealing. Maybe explain.

---

> ### Author Response · Authors · 2025-11-21
>
> We thank the reviewer for their feedback.
>
> **Query**: Overall I am still hungry after reading the paper. From what I have seen in the algorithm community, is that most hard instances consists of some fractions, to which some epsilon values have been added or removed.
>
> **Response**:
> We think that the statement about the nature of lower bound constructions is oversimplified. While for some algorithms one can indeed find structurally simple lower bounds that consist of some fractions, to which some epsilon values have been added or removed, this is not a general rule. Such simple lower bound examples can sometimes be found in textbooks but this selection is biased because textbooks usually prioritize simple constructions. Lower bound constructions in advanced algorithmic research are often much more complex and harder to find. See, e.g., https://arxiv.org/abs/2407.20422 for a lower bound on the greedy algorithms for the Shortest Common Superstring Problem or lower bounds for the price of hierarchy https://doi.org/10.1007/s00453-025-01327-7 These are just two of very many examples.
> It is not clear why this is a weakness of our results.
>
> **Query**:Also I think the approach could benefit from the design of a grammar describing compactly instances, rather than generating a Python program, because the search space is more targeted to a form of aimed instances.
>
> **Response**:We conducted preliminary experiments utilizing a custom grammar as an alternative to generating Python code, exploring both fine-tuning LLMs for this grammar and performing direct searches within the grammar space. However, we observed that LLMs leverage their extensive pre-training on internet-scale corpora to incorporate relevant algorithmic knowledge more effectively than blind searches in a custom grammar space. Furthermore, current models demonstrate significantly higher efficiency and proficiency in high-level languages like Python and C++, attributable to the massive representation of these languages in their pre-training data."
>
> **Query**: You don't give the evaluation function to the LLM. Explain this choice.
>
> **Response**: Thanks for the suggestion. Not including the evaluation-function in the prompt was the default choice, made by the previous work on FunSearch (Romera-Paredes et al, 2024).  See https://github.com/google-deepmind/funsearch/blob/cc53f274237d7ab05c19df939edbc1f9616a7c19/implementation/programs_database.py#L236
>
> Nevertheless, on your suggestion, we re-ran the Nemhauser-Ullmann FunSearch with the evaluation-function within the prompt. Adding the evaluation-function in the prompt does seem to yield better results. The standard-error is large, despite running 30 trials each.
> Counting up the tokens, “with evaluation function” uses about 2.6 times as many input-tokens as “without-evaluation”, and about 1.6 times as many output-tokens as “without-eval”. Accounting for input-tokens being cheaper than output-tokens (especially because we can batch input-tokens), this means that the “with-eval” trial was about 1.6 times as expensive as the without-eval trial. So, there is a cost-performance trade-off as well in using evaluation function in the prompt or not.
> We have added a plot in the appendix for the same in the updated submission.
>
> **Query**: Line 372, explain what each dimension means. I guess first is weight, and second is profit. Several times you mention a choice of temperature, which I only know in the context of simulated annealing. Maybe explain.
>
> **Response**:  We write in line 366 in original submission (line 377 in modified versions)that items are written as tuples (weight, profit). The 'temperature' mentioned here refers to the scaling parameter $T$ applied to the output logits prior to the softmax operation, which regulates the entropy of the probability distribution. This is commonly referred to as “sampling temperature” in the LLM community.

---

### Official Review · Reviewer_D4Qs · 2025-11-01

**Soundness:** 3
**Presentation:** 3
**Contribution:** 3
**Rating:** 6
**Confidence:** 3

**Summary:**

The paper proposes a method for finding adversarial instances for combinatorial optimization heuristics based on FunSearch. They use LLMs for creating problem instances in a form of Python programs where given heuristic does not perform well. The authors argue that this representation is more interpretable than less structured instance vectors obtained for example by local search. The paper focuses on heuristics for four combinatorial optimization problems including the knapsack problem, bin packing, hierarchical clustering and a variant of the gasoline problem. The authors find tighter lower bounds than the ones that were known before for these problems using their method of Co-FunSearch.

**Strengths:**

1. Originality

The paper is based upon existing FunSearch technique but it introduces novel elements such as including experts in the search pipeline in their Co-FunSearch variant. They also apply FunSearch to the combinatorial optimization domain. The authors also find novel theoretical results for the considered problems which seem to be not have been known before this work.

2. Quality

The paper presents numerous experiments and tests proposed methodology for four important problems thus providing a solid experimental base for their results.

3. Clarity

The paper is well-written and easy to follow.

4. Significance

Combinatorial optimization heuristics have important practical and theoretical meaning. Constructing adversarial examples for them provides additional insights into the nature of the problems for which they were created. Using an interpretable representation of such adversarial examples such as Python-code can be useful for further analysis of these examples and constructing novel heuristics which would be more robust.

**Weaknesses:**

1. (Significance) Co-FunSearch relies on collaboration with human experts which might be a bottleneck for large-scale applications of the method. As the authors state (lines 076-077), expert modifications were essential for generating meaningful insights. Thus, the methodology can be used as a supporting tool but not as a standalone process for generating adversarial examples.

2. (Clarity) Even though local search is known in the field of combinatorial optimization, could you specify in more detail how exactly you perform it since it is one of the baselines in your experiments.

**Questions:**

1. Can Co-FunSearch be made fully autonomous by using agents for refining found programs (lines 076-077, Figure 1c, 3c)?

2. The authors argue that solutions found by their method are more interpretable and symmetric when compared to solutions found by local search (lines 066-067). It would be interesting to see whether they are also more robust to noise. For example, if we take a vector provided by local search such as the one in the footnote 1 (lines 106-107) and add some small noise to its element, will it maintain it's properties as an adversarial example for a given heuristic? What if we do the same to solutions provided by your method?

---

> ### Author Response · Authors · 2025-11-21
>
> We thank the reviewer for the encouragement and feedback.
>
>
>
>
>
> **Query**: (Significance) Co-FunSearch relies on collaboration with human experts which might be a bottleneck for large-scale applications of the method. As the authors state (lines 076-077), expert modifications were essential for generating meaningful insights. Thus, the methodology can be used as a supporting tool but not as a standalone process for generating adversarial examples.
>
> **Response**: While we agree that human expert refinement is an important component of the Co-FunSearch framework, this is a common trend in obtaining state-of-the-art in recent works: (a) Terrence Tao’s latest publication on Nikodym set constructions (https://arxiv.org/abs/2511.07721) (b) Advancing mathematics by guiding human intuition with AI (https://www.nature.com/articles/s41586-021-04086-x). Current AI systems lack the capacity to autonomously resolve complex open problems in theoretical computer science and mathematics. These papers and ours suggest a collaboration between experts and AI offer complementary advantages for solving such challenges.
>
>
> **Query**:(Clarity) Even though local search is known in the field of combinatorial optimization, could you specify in more detail how exactly you perform it since it is one of the baselines in your experiments.
>
> **Response**: These details can be found at the beginning of section 3 (starting at line 214 in original submission and line 225 in modified version). We start with a random instance, encoded as a vector. At each step, we perturb this vector by adding normally-distributed noise to it, with variance decaying over time. If this perturbed vector has a higher objective than the previous vector, we keep the perturbed vector, and otherwise we keep the previous vector.
>
>
> **Query**: Can Co-FunSearch be made fully autonomous by using agents for refining found programs (lines 076-077, Figure 1c, 3c)?
>
>
> **Response**: "We believe that current AI models are insufficient for the autonomous refinement proposed by Co-FunSearch. Although agents can handle syntactic simplification, challenges persist in instance analysis and formal verification. Addressing these gaps through the integration of proof agents is an interesting direction for future research."
>
> **Query**: The authors argue that solutions found by their method are more interpretable and symmetric when compared to solutions found by local search (lines 066-067). It would be interesting to see whether they are also more robust to noise. For example, if we take a vector provided by local search such as the one in the footnote 1 (lines 106-107) and add some small noise to its element, will it maintain it's properties as an adversarial example for a given heuristic? What if we do the same to solutions provided by your method?
>
> **Response**:   As described in the paper, funsearch prefers structured and symmetric instances and is thus not expected to be robust to noise. The randomness would break the structure and symmetry of the instances. In any case, resistance to noise is not the focus of our work.
> The instances found by local search could be more robust to noise but are definitely not generalizable as highlighted in the example.

---

### Official Review · Reviewer_zZp2 · 2025-11-04

**Soundness:** 2
**Presentation:** 2
**Contribution:** 2
**Rating:** 2
**Confidence:** 4

**Summary:**

This paper extends FunSearch to generate adversarial examples for heuristics for combinatorial optimization. Specifically, the authors look at knapsack, bin packing, k-median clustering, and the gasoline problem. The value of the task is generating practical samples that are hard to approximate for a certain solver, closing the gap to the worst-case upper bound.

**Strengths:**

* Using LLM to tackle math and optimization problems is a trending research topic.
* The proposed Co-FunSearch framework seems to outperform FunSearch in the experimental evaluations.
* An ablation study is provided in the experiment section.

**Weaknesses:**

* While the idea of identifying worst-case instances for existing heuristics is conceptually interesting, it is unclear whether this direction achieves the same level of impact or technical novelty as improving the heuristics themselves — as demonstrated, for example, in FunSearch and its case studies.

* Significant revisions in writing and presentation are necessary before the technical contributions of this work can be properly evaluated.
    * The introductory paragraph discussing AI advancements in biology, chemistry, and mathematics is tangential to the main topic and should be removed to improve focus.
    * A substantial portion of the paper is devoted to describing problem definitions, heuristics, and code snippets, yet key implementation details of the proposed method are missing. It remains unclear how Co-FunSearch differs from FunSearch, or how it can be concretely instantiated with a given heuristic. The authors should include algorithmic diagrams or pseudo-code in a dedicated “Method” section to clarify these aspects.
    * If the authors wish to elaborate on problem formulations, additional context and explanations are needed for readers unfamiliar with the examples. For instance, the so-called “famous gasoline problem” attributed to Lovász is insufficiently explained; as presented, it is not interpretable and unlikely to be “famous” to the broader audience.
    * In line 214, the authors state:
      > “The main goal in all these problems is to search for a vector v which optimizes the given objective.”

      This statement lacks precision. It is unclear
        1. what the “vector v” represents and how it fits within the different search paradigms described, and
        2. what the “given objective” is, i.e., which function or metric is actually being optimized.

**Questions:**

* In the footnote on page 2,
    > For instance, one of the local-search-generated lists outperforming FunSearch was: [0.003031, 0.005466,
0.006098, 0.007283, 0.021158, 0.068030, 0.073417, 0.170490, 0.202092, 0.219287, 0.306771, 0.375912,
0.540358].

    It is unclear how this is relevant to a "discernible pattern" and please provide explanations.
* To compute the approximate ratio, the optimal value is required. How do you know the optimal value if the problem parameter is always changing? Is the scalability of this framework bottlenecked by solving for the optimal value?
* How many problem instances are used to calculate Table 1? How are they collected? How do you determine the size of the problem instance?
* In Table 1, what does "Previous Best Lower Bound" mean?
* In Table 1, what is the "Best-Fit" problem?

---

> ### Author Response · Authors · 2025-11-21
>
> We thank the reviewer for the feedback.
>
> **Query**: While the idea of identifying worst-case instances for existing heuristics is conceptually interesting, it is unclear whether this direction achieves the same level of impact or technical novelty as improving the heuristics themselves — as demonstrated, for example, in FunSearch and its case studies.”
>
> **Response**: Finding lower bounds for algorithms is an integral part of algorithmic research and usually goes hand in hand with developing better algorithms. For example, when developing new approximation algorithms for a problem, one is often faced with the situation that one has developed an algorithm that one suspects to achieve a certain approximation factor. Then the typical approach of an algorithms' researcher is to try simultaneously to prove an upper bound and to find a lower bound. While arguably, finding an upper bound is usually the more pleasant outcome, also lower bounds give valuable insights because they show that the algorithm needs to be adapted to achieve a better upper bound and sometimes they even show how the algorithm needs to be modified. The same reasoning applies also to the analysis of the running time of an algorithm instead of the approximation factor.
>
> One of very many prominent prototypical examples where researchers have devoted a significant amount of time (decades) on finding lower bounds for algorithms is the Greedy algorithm for the Shortest Common Superstring Problem: https://arxiv.org/abs/2407.20422 Other important examples include lower bounds for the running time of the policy iteration algorithm for MDPs (https://arxiv.org/abs/2506.12254) and lower bounds for various pivot rules for the simplex algorithm (https://link.springer.com/chapter/10.1007/978-3-031-93112-3_7).
>
> We adopt in our submission the classical worst-case perspective from theoretical computer science. Indeed this implies that lower bound examples are sometimes contrived and do not always reflect typical inputs. This problem is addressed by other models like smoothed analysis but worst-case analysis is still the dominant measure in the analysis of algorithms. So we believe that our work is meaningful and adapting it to other models like smoothed analysis is an interesting topic for future research.
>
> **Query**: The introductory paragraph discussing AI advancements in biology, chemistry, and mathematics is tangential to the main topic and should be removed to improve focus.
>
> **Response**: Thanks for the feedback! We edited the paragraph accordingly.
>
> **Query**: A substantial portion of the paper is devoted to describing problem definitions, heuristics, and code snippets, yet key implementation details of the proposed method are missing. It remains unclear how Co-FunSearch differs from FunSearch, or how it can be concretely instantiated with a given heuristic. The authors should include algorithmic diagrams or pseudo-code in a dedicated “Method” section to clarify these aspects.
>
> **Response**: The entire methodology that will serve an answer to this query is explained in lines 69-79 in the original submission (lines 75-86 in modified version). We clarify that our primary contribution is a collaboration between the FunSearch method and experts to solve a diverse set of problems. We illustrate how FunSearch can serve as an effective tool for establishing new lower bounds.
> We have added an algorithmic diagram to illustrate the Co-FunSearch workflow for further clarity.
>
> **Query**: If the authors wish to elaborate on problem formulations, additional context and explanations are needed for readers unfamiliar with the examples. For instance, the so-called “famous gasoline problem” attributed to Lovász is insufficiently explained; as presented, it is not interpretable and unlikely to be “famous” to the broader audience.
>
> **Response**: Thank you! We will rephrase.
>
> **Query**: In line 214, the authors state:
> “The main goal in all these problems is to search for a vector v which optimizes the given objective.”
> This statement lacks precision. It is unclear
> what the “vector v” represents and how it fits within the different search paradigms described, and
> what the “given objective” is, i.e., which function or metric is actually being optimized.
>
> **Response**: The vector v encodes the instance of the problem. For the bin-packing problem with capacity 1 and n items, this could be a vector v in [0, 1]^n, where the i-th entry of v is the weight of the ith item. The objective is usually the performance-measure of the respective algorithm. For instance, for the randomised best-fit problem, the objective is to find an instance where the expected output of randomised best-fit uses a large number of bins when compared to an optimal solution. The specific objective for each problem is detailed in section 3.3.

---

> > ### Author Response · Authors · 2025-11-21
> >
> > **Query**: In the footnote on page 2,
> > For instance, one of the local-search-generated lists outperforming FunSearch was: [0.003031, 0.005466, 0.006098, 0.007283, 0.021158, 0.068030, 0.073417, 0.170490, 0.202092, 0.219287, 0.306771, 0.375912, 0.540358].
> > It is unclear how this is relevant to a "discernible pattern" and please provide explanations.
> >
> > **Response**: The main objective of this example is exactly that an instance found by local search is not interpretable. Although some of these instances could achieve a high score, no logical insight could be discovered from these instances by experts, hence they are not generalizable. On the other hand, the instances found by FunSearch are described in short code. Thus, most of these instances have a logical structure and can be improved by experts to obtain new provable insights.
> >
> > **Query**: To compute the approximate ratio, the optimal value is required. How do you know the optimal value if the problem parameter is always changing? Is the scalability of this framework bottlenecked by solving for the optimal value?
> >
> > **Response**: Yes, the optimal value is required for solving approximate ratio and we attempt to get this optimal by running a relevant solver like ILP solver (like Gurobi), a packing-solver [Fontan & Libralesso], or a best-first-search solver we wrote ourselves. Moreover, it should be noted that in these runs, even a lower bound is sufficient enough for optimising with funsearch and an optimal value is not required.
> > (Descriptions on how we computed the optimum values are also present in the beginning of each section 3.3.1 to 3.3.4.)
> > We did calculate the optimal value exactly. For large problems this would become intractable, but for the small problems we considered, it was on par with the time it took to query the LLM. For more difficult or larger problems, one could also compute an upper bound on the optimal solution, which can usually be done much more quickly, but this would only provide a lower bound on the score of each instance.
> >
> > **Query**: How many problem instances are used to calculate Table 1? How are they collected? How do you determine the size of the problem instance?
> >
> > **Response**: For local-search, the sizes of the local-search instances are given at the beginning of section 3 (starting in line 222 in original submission and line 232 in modified submission). These sizes were chosen manually to strike a balance between speed of computation (larger instances will quickly become intractable to solve), while still being large enough to achieve high scores. For this, we also considered the sizes of the instances found by FunSearch.
> > For FunSearch and every problem except the gasoline-problem, the size of the instance was not specified; the program could return an instance of any size. However, the size of the instances was indirectly limited by a computation-timeout of 10 seconds. If the evaluation of a program exceeded 10 seconds, we considered it intractable and assigned a score of 0.
> > For the gasoline-problem, the program was parametrised over the size of the instance, and evaluated for n ∈ {6, 14, 30}, which were the relevant sizes for the previously known lower-bound construction.
> > To calculate Table 1, we ran 30 trials for each problem.
> >
> > **Query**: In Table 1, what does "Previous Best Lower Bound" mean?
> >
> > **Response**: These are the best lower bounds that were previously known for each problem, and our work provided better (higher) lower bounds for the listed problems.
> >
> > **Query**: In Table 1, what is the "Best-Fit" problem?
> >
> > **Response**: Best-Fit is a heuristic for solving the Bin-Packing-Problem. In the paper, we considered the performance-ratio of Best-Fit on shuffled worst-case instances, when compared to an optimal solution. We renamed the column to “Bin-Packing” for consistency. Thank you!

---

### Note · Authors · 2026-01-19

I have read and agree with the venue's withdrawal policy on behalf of myself and my co-authors.